# AbsenceBench: Language Models Can't Tell What's Missing

**Harvey Yiyun Fu**[*,1], **Aryan Shrivastava**[1], **Jared Moore**[2]
**Peter West**[2], **Chenhao Tan**[1], **Ari Holtzman**[1]
[1]University of Chicago   [2]Stanford University

## Abstract

Large language models (LLMs) are increasingly capable of processing long inputs and locating specific information within them, as evidenced by their performance on the Needle in a Haystack (NIAH) test. However, while models excel at recalling surprising information, they still struggle to identify *clearly omitted* information. We introduce `AbsenceBench` to assesses LLMs' capacity to detect missing information across three domains: numerical sequences, poetry, and GitHub pull requests. `AbsenceBench` asks models to identify which pieces of a document were deliberately removed, given access to both the original and edited contexts. Despite the apparent straightforwardness of these tasks, our experiments reveal that even state-of-the-art models like Claude-3.7-Sonnet achieve only 69.6% F1-score with a modest average context length of 5K tokens. Our analysis suggests this poor performance stems from a fundamental limitation: Transformer attention mechanisms cannot easily attend to "gaps" in documents since these absences don't correspond to any specific keys that can be attended to. Overall, our results and analysis provide a case study of the close proximity of tasks where models are already superhuman (NIAH) and tasks where models breakdown unexpectedly (`AbsenceBench`).

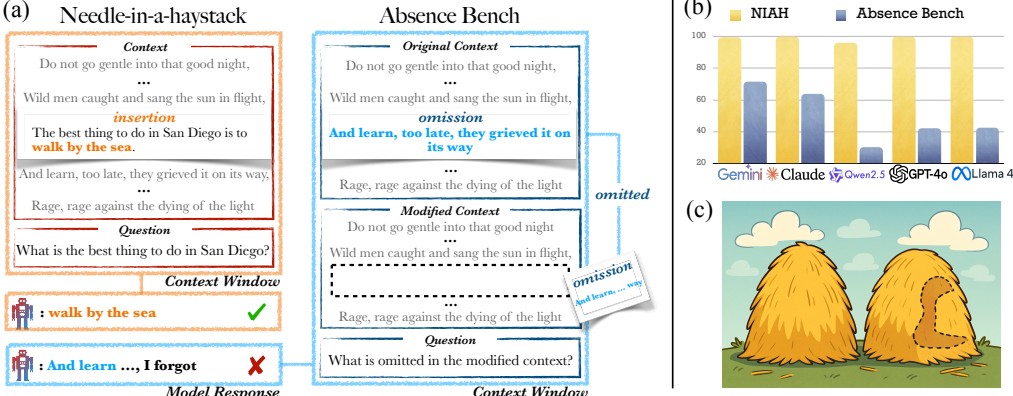

Figure 1: (a) An overview of the difference between the Needle-in-a-haystack (NIAH) test setting and `AbsenceBench` task setting. `AbsenceBench` is asking models to identify omitted pieces of content. (b) Performance of 5 SoTA LLMs on `AbsenceBench` is significantly lower than on the NIAH test, measured by F1-score. (c) an illustration of our task setting using the "haystack" metaphor, generated by ChatGPT.

---
[*]*Corresponding author(s):* harveyfu@uchicago.edu

39th Conference on Neural Information Processing Systems (NeurIPS 2025) Track on Datasets and Benchmarks.

# 1 Introduction

Recent Large Language Models (LLMs) have shown exceptional abilities across a wide range of long-context tasks [Bai et al., 2024, Zhang et al., 2024, *inter alia*]. In particular, the Needle-in-a-Haystack (NIAH) test [Kamradt, 2023, Hsieh et al., 2024] has been used to evaluate whether models can find small bits of surprising information in extremely long inputs, with recent models pushing the frontier of NIAH into the millions of tokens. We ask the converse question: **Can LLMs spot information that has clearly been left out?**

To answer this question, we present `AbsenceBench`: a new benchmark designed to evaluate the abilities of LLMs in locating conspicuously missing information from long inputs. Instead of asking LLMs to find off-topic information (the 'needle' in NIAH), LLMs are prompted to identify and recall intentionally **omitted** information. Cutting-edge, closed-source models perform poorly on `AbsenceBench`, despite the apparent similarity to NIAH and the fact that `AbsenceBench` is simple to describe and entirely unambiguous.

`AbsenceBench` includes three domains: poetry, numerical sequences, and GitHub pull requests (PRs). The average context length of `AbsenceBench` is 5K, significantly shorter than most long-context benchmarks—we thus call it a medium-context benchmark. We benchmark a total of 14 LLMs on `AbsenceBench`, including cutting-edge models such as GPT-4 [OpenAI et al., 2024b], Claude-3.7-Sonnet [Anthropic, 2024], and Gemini-2.5-flash [Google, 2025][2], as well as inference-time compute models such as o3-mini [OpenAI, 2025], Grok-3-mini [xAI, 2025] and DeepSeek-R1 [DeepSeek-AI et al., 2025]. We further study the impact of context length and the rate of omission. Our results suggest that: (1) Longer context length makes the task harder especially under the poetry domain. (2) Using inference-time compute boost models' performance by only 7.9% at the cost of generating an average of extra 8K thinking tokens—nearly 3x the average document length! (3) Counter-intuitively, a *lower* rate of omission in the task set-up leads to *worse* model performance.

`AbsenceBench` is also significantly more difficult for LLMs than the NIAH test. To illustrate this, we compare the performance of three LLMs on both our proposed task setting and the original NIAH test setting, observing a a massive *56.9%* drop in F1-score on average. Why is `AbsenceBench` so much more difficult than the NIAH test? We hypothesize that the Transformer [Vaswani et al., 2017] attention may have trouble addressing "gaps" in the document, leading us to experiment with adding placeholder strings where information is omitted. This boosts the performance by a dramatic 35.7% on average (see §4.2).

What explains these results? While LLMs often generate deceptively human-like responses, and may even have human-like cognitive-biases [Koo et al., 2024], the contrast in success between NIAH and `AbsenceBench` suggests that models may have failed for completely different reasons than humans do. This has practical implications: if LLMs cannot properly notice when information is missing, can we rely on LLM-as-a-Judge [Zheng et al., 2023] for a wide range of tasks where recognizing missing information matters? We hope that `AbsenceBench` can serve as a starting point for examining LLMs' ability to tell what is missing, and as a case-study of an LLM-specific cognitive bias.

Overall, our contributions are as follows:

- We release a new medium-length-context benchmark for evaluating LLMs' abilities in locating intentionally-omitted information across three diverse domains.
- We evaluate 14 popular LLMs, including those with inference-time compute, and show that our benchmark is challenging even for cutting-edge language models.
- We show that while the the NIAH test is essentially solved for very long contexts, `AbsenceBench` tasks have low performance on medium length contexts, suggesting LLMs find it harder to identify omissions than insertions.
- We analyze the effect of inference-time compute, finding that it offers only a modest performance improvement at a high cost—an average chain-of-thought length that is nearly 3x the size of the average document length.
- Explicitly marking omissions improves models' performance by 35.7% on average. This suggests that `AbsenceBench` may be caused by inherent weaknesses within Transformer-style self-attention mechanisms.

---

[2]Claude-3.7-Sonnet and Gemini-2.5-flash have the option to leverage inference-time compute and we evaluate them both with and without this feature.

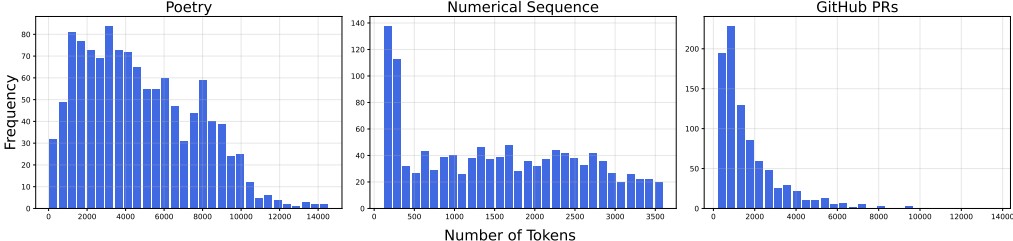

Figure 2: **The three domains in `AbsenceBench` test models' abilities across a variety of document lengths and omission probabilities**. Frequency reports the number of tasks in the domain within a given range of document lengths. The average context length across all tasks in `AbsenceBench` is 5K tokens. On the document level, the average document length is 2.7K, while it is 4.7K for poetry, 1.5K for numerical sequences, and 1.7K for Github pull requests. We use the GPT-4 Tokenizer[4] to measure document and context lengths.

| Domain | Original | Modified |
|--------|----------|----------|
| Poetry | . . .And so, to you, who always were
To me, I give these weedy rhymes
In memory of early times. . . | . . .And so, to you, who always were
In memory of early times. . . |
| Numerical | 117, 121 , 125, 129, 133, 137 . . . | 117, 125, 129, 133 . . . |
| GitHub PRs | . . .
+  \$replacements = [
+    '[' => '\[',
+    '<' => '<',
+    '>' => '>',
+  ]; | . . .
+  \$replacements = [
+    '[' => '\[',
+    '>' => '>',
+  ]; |
| Absence Token | . . .And so, to you, who always were
To me, I give these weedy rhymes
In memory of early times. . . | . . .And so, to you, who always were
<missing line>
In memory of early times. . . |

Table 1: Examples of tasks in `AbsenceBench` by domain. Models are given both the original and modified documents and are asked to identify which elements are missing (in blue ). The benchmark includes the Poetry, Numerical Sequences, and Github PRs domains, while the Absence Token domain includes an *omission token* that is used for analysis in §4.2

# 2  `AbsenceBench`

`AbsenceBench`[3] is built on a simple idea: LLMs have trouble keeping track of what's missing.

## 2.1  Task Definition

We formulate all tasks in `AbsenceBench` into a straightforward controlled generation framework: given an original document composed of $n$ elements $D_{\mathrm{orig}} = \{e_1, \ldots, e_n\}$, we intentionally omit $p$ percent of the elements $D_{\mathrm{omit}} = \{e_{r1}, e_{r2}, \ldots, e_{rk}\}$, with $k = p \cdot n$, and produce a modified version of the document $D_{\mathrm{modified}} = D_{\mathrm{orig}} \backslash D_{\mathrm{omit}}$. We present both versions of the document $D_{\mathrm{orig}}$ and $D_{\mathrm{modified}}$ to the LLMs and then prompt them to generate the exact set of omitted elements $D_{\mathrm{omit}}$. We use "document length" to denote the token count of the original document, and "context length" to indicate the token count of the entire input context.

## 2.2  Dataset Construction

---

[3]Our code is available at `https://github.com/harvey-fin/absence-bench`
[4]`https://github.com/openai/tiktoken?tab=readme-ov-file#-tiktoken`

Table 2: Default prompt template used for evaluating language models under the poetry domain

AbsenceBench covers three distinct domains: poetry, numerical sequences, and GitHub pull requests (PRs). Poetry and GitHub PRs are realistic data directly collected from open sources, while numerical sequences are synthetic. We choose $p = 0.1$ for all domains. Overall, AbsenceBench contains 4302 instances in total, with an average context length of 5K tokens. We plot the distribution of document lengths for each domain in Figure 2. We include justifications for our construction parameter choices in Appendix D.

**Poetry**   We use poems from the Gutenberg Poetry Corpus [Parrish, 2018]. For each poem, we omit at a line level and use the newline character as the delimiter between different lines. In order to create diversity in document length, we truncate the poems such that the number of lines for each poem are uniformly distributed from 100 to 1000. We include justification over the choices of dataset construction parameters in Appendix

**Numerical Sequences**   LLMs demonstrate impressive abilities in solving mathematical tasks [Frieder et al., 2023], and have seen numbers frequently in the pre-training data. Nonetheless, it is unclear whether LLMs are able to reliably keep track of numerical sequences. We generate a total of 1200 numerical sequences. Each sequence consists of $n$ decimal numbers $\{a^{(1)}, a^{(2)}, \ldots, a^{(n)}\}$ that are arranged in a particular order $\mathcal{L} \in \{\text{ascending, descending, random}\}$, with a step size $s \in \{1, 4, 7, 13\}$ between two consecutive numbers. The first number $a^{(1)}$ is randomly chosen from 0 to 9999. We omit each number from the set with a certain probability, and delimit numbers with a newline character.

**GitHub PRs**   GitHub is one of the largest open-source code repositories and is frequently used as a high-quality data source for (LLMs) such as Code Llama [Rozière et al., 2024]. We retrieve the PRs from the top 20 GitHub repositories[5] with the most PRs using GitHub API, and only keep those PRs with a diff that includes 10 to 200 updated lines (i.e., a line that starts with "+" or "-"). We format this task similarly to the poetry domain: omissions are applied at a line level, and we use the newline character as the delimiter. In particular, we only omit the updated lines that are unique within each PR's diff. This domain has practical application to current LLM-usage: an LLM that resolve and verify merge conflicts must inherently be able to detect omissions in file diffs.

## 3 Experiments

### 3.1 Experiment Setup

**Models**   We evaluate a total of 14 LLMs, including seven models that leverage inference-time compute (Gemini-2.5-flash [Google, 2025], Claude-3.7-Sonnet [Anthropic, 2024], o3-mini [OpenAI, 2025], DeepSeek R1 [DeepSeek-AI et al., 2025], Grok-3-mini-Beta [xAI, 2025], Qwen3-235B [Qwen, 2025a], QwQ-32B [Qwen, 2025b]), three OpenAI models (GPT4o[OpenAI et al., 2024b], GPT-4.1, GPT-4.1-mini), and four open-weights models (Llama-4-Maverick, Llama-3.3-70B-Instruct [Meta, 2025], Qwen2.5-72B-Instruct [Yang et al., 2024], Mixtral-8x7B-Instruct [Jiang et al., 2024]). We run both Gemini-2.5-flash and Claude-3.7-Sonnet with and without inference-time compute (i.e.,

---

[5] https://top1000repos.com/based-on-pr

| Models | Poetry | Numerical Sequences | GitHub PRs | Average |
|---|---|---|---|---|
| Gemini-2.5-flash* | **87.3** | 95.4 | 30.9 | **71.2** |
| Claude-3.7-Sonnet* | 72.7 | **96.0** | **40.0** | 69.6 |
| Claude-3.7-Sonnet | 73.5 | 91.4 | 35.7 | 66.9 |
| Gemini-2.5-flash | 79.3 | 85.2 | 26.2 | 63.6 |
| o3-mini* | 65.0 | 78.1 | 38.9 | 60.7 |
| GPT-4.1 | 54.3 | 57.5 | 36.2 | 49.3 |
| Grok-3-mini-Beta* | 40.7 | 56.3 | 36.4 | 44.5 |
| GPT-4o | 38.4 | 48.1 | 39.4 | 42.0 |
| QwQ-32B* | 32.1 | 57.7 | 31.6 | 40.5 |
| Llama-4-Maverick | 32.8 | 58.7 | 29.0 | 40.2 |
| GPT-4.1-mini | 30.2 | 45.0 | 31.3 | 35.5 |
| Llama-3.3-70B-Instruct | 25.3 | 37.7 | 28.7 | 30.6 |
| Qwen2.5-72B-Instruct | 19.0 | 45.4 | 26.8 | 30.4 |
| DeepSeek-R1* | 38.7 | 29.5 | 23.1 | 30.4 |
| Qwen3-235B* | 26.1 | 18.5 | 24.6 | 23.1 |
| Llama-3.1-8B-Instruct | 9.4 | 17.1 | 21.6 | 16.0 |
| Mixtral-8x7B-Instruct | 4.9 | 21.9 | 17.3 | 14.7 |

Table 3: Micro F1-score (%) of 15 LLMs evaluated on all three domains of Absence Bench. * indicates the model uses inference-time compute during evaluation. GitHub PRs represent the most challenging domain: models that perform well in other domains often struggle there.

"thinking mode"). All selected models have a claimed context length of over 32K tokens. All model inferences are obtained through API requests (see Appendix B for details).

**Evaluation Metric** Our evaluation is two-fold: (1) we use exact match to check whether each element (e.g., a line of a poem) is present in a model's response, and (2) we use **micro F1-score** to evaluate whether models generate the correct set of case-insensitive elements. Note that we use a different evaluation metric than the recall accuracy metric that is frequently used in the NIAH test. In our task setting, a model could achieve a perfect recall score simply by copy-pasting the entire original context. However, this would result in many false positives cases (i.e., when model generates non-omitted elements) that recall accuracy does not account for.

**Prompt Templates & In-context Learning** Table 2 presents an example of the prompt template used for evaluation in the poetry domain. See Appendix A for detailed prompts under all domains. Due to compute and context length constraints, we include limited perturbation studies on the prompt template and in-context learning examples in this paper and leave it for future work. See Appendix C for a perturbation study on the prompt template and §7 for further discussion.

## 3.2 Main Results

We show the evaluation results of 14 LLMs on `AbsenceBench` in Table 3. Gemini-2.5-flash (thinking)[6] outperforms the rest of the models on poetry, numerical sequences, and overall by a significant margin, followed by Claude-3.7-Sonnet (thinking). Most open-weights models are generally weaker in performance and struggle to reach 40% F1-score except for QwQ-32B and Llama-4-Maverick. GitHub PRs is a challenging domain for all models: the highest score is only 40.0% by Claude-3.7-Sonnet (thinking). It is important to highlight that while Mixtral-8x7B achieves a perfect score on the NIAH test at an equivalent context length, it attains only a 14.7% F1-score on the `AbsenceBench`, indicating substantial space for improvement among smaller-scale models. Overall, `AbsenceBench` presents a surprisingly challenging task to cutting-edge language models.

**Inference-time compute is helpful but costly** We consider seven models that include inference-time compute capabilities. We plot the distribution of *thinking token ratio*: the ratio between the total thinking tokens generated by seven inference-time compute models and the original document length under all three domains in Figure 3. The figure shows that reasoning models typically generate thinking tokens several times greater than the document length, which further suggests that the models

---

[6]We use (thinking) to denote the model that use inference-time compute during evaluation

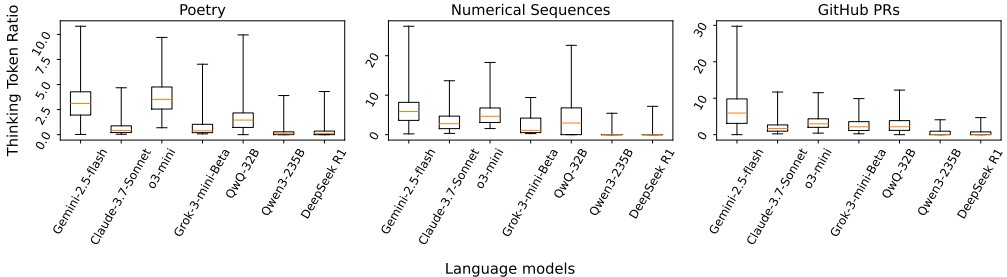

Figure 3: **Reasoning models often generate an order of magnitude more text than input document.** Distribution of the *thinking token ratio* (number of generated thinking tokens divided by number of tokens in the original document) for four inference-time compute models under each domain. We set the parameters of the boxplot to capture 99% of the distribution. The outliers are hidden for better clarity (see Figure 8 for the full distribution).

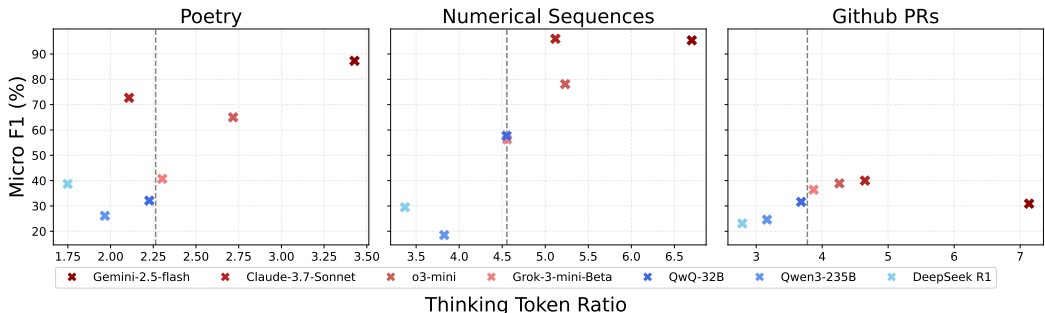

Figure 4: Closed-source models (reds) perform better than open-weights models (blues) on `AbsenceBench`, while generating more thinking tokens. Each plot shows the average F1-score (x-axis) and the average thinking token ratio (y-axis). The grey line presents a visual boundary.

may attempt to reconstruct the original document using thinking tokens as a way to identify omissions. Notably, Gemini-2.5-flash shows the highest thinking token ratio of 5.7 on average across three domains, partly accounting for its advantage in benchmark performance. Additionally, we compare the performance of Gemini-2.5-flash and Claude-3.7-Sonnet under conditions with and without the use of "thinking mode". We observe that inference-time compute leads to a modest performance improvement of 7.9%, but it comes at the cost of producing an additional 8K tokens for intermediate reasoning steps on average—significantly longer than our average context length of 5K tokens.

**Closed-source models outperform open-weights models**   We plot the average F1-score and thinking token ratio of four closed-source models and three open-weights models in Figure 4. We observe that closed-source models generally achieve higher average F1-scores than open-weight models across all domains, with the exception of QwQ-32B, which outperforms Grok-3-mini-Beta in numerical sequences and Gemini-2.5-flash in GitHub PRs. On the other hand, the higher performance of closed-source models come at the cost of generating 42.0% more thinking tokens than open-weights models on average across all domains.

**Locating omissions are harder than insertions**   The increased difficulty of `AbsenceBench` relative to the original NIAH test could potentially be attributed to differences in the task design and the higher frequency of document modifications (insertions or omissions). To investigate this, we run Claude-3.7-Sonnet, GPT-4.1-mini, and Llama-4-Maverick under an "insertion bench" setting: rather than omitting elements from the context, we insert "needles" at an equivalent frequency. We choose the Harry Potter dataset[7] as the needles and two realistic domains (poetry and GitHub PRs) as the haystack. All three models achieve nearly 99.5% F1-score on the poetry domain, and at least 86.2% F1-score on Github PRs (see Appendix E for full results). By comparison, models

---
[7] https://www.kaggle.com/datasets/gulsahdemiryurek/harry-potter-dataset

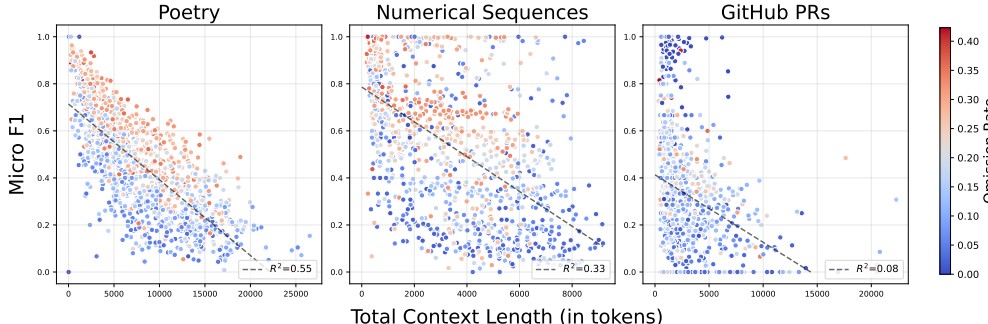

Figure 5: **GPT-4.1-mini performs worse on longer tasks in Poetry,** but the relationship is not clear in Numerical Sequences and Github PRs. Each plot shows the F1-score (y-axis) and the total context length (x-axis). Dark blue represents a lower and dark red represents a higher percentage of omission (number of omitted lines divided by total number of lines across each of three domains). Each dot on the graph represents the performance on a single instance. The dashed lines represent the least squares fit, with $R^2$ indicating the strength of correlation between the two axes.

experience a substantial average performance drop of 56.9% under the `AbsenceBench` task setting, suggesting that the observed difficulty likely arises from the distinction between omissions and insertions. A possible explanation for this observation is that the Transformer attention mechanism might struggle to effectively handle "gaps" when processing documents containing omissions. We carry out additional experiments to assess this hypothesis in Section 4.2.

# 4 Analysis

We study the effect of perturbing the context length as well as the percentage of omissions on the models' performance. We additionally perform an error analysis and present a simple strategy that effectively addresses the difficulty in `AbsenceBench`. For the sake of cost, we limit our extended analysis to three LLMs: Claude-3.7-Sonnet, GPT-4.1-mini, and Llama-4-Maverick.

## 4.1 Perturbation Studies

We construct a perturbed dataset with a higher variance of omission rates in order to lower the impact of various factors on the difficulty of `AbsenceBench` tasks. We randomly sample the probability of omission to be $p \sim U(0, 0.5)$ for each document, and then randomly omit each element from that document with probability $p$. Figure 5 illustrates the relationship between the performance of GPT-4.1-mini and total context length. The analysis of Claude-3.7-Sonnet and Llama-4-Maverick are shown in Figure 6 and 7 in Appendix F.

**Context length has a major impact on performance**  For contexts shorter than 5K tokens, the performance varies significantly with changes in context length. This finding contrasts with observations from the NIAH test, where context length affects performance only when contexts exceed 100K tokens. We perform a linear regression analysis to assess the correlation between F1-score and total context length. Results across three LLMs indicate a stronger average correlation (R² = 0.55) within the poetry domain compared to numerical sequences (0.19) and GitHub PRs (0.08).

**LLMs perform worse with lower omission rate**  In Figure 5, we observe a notable pattern (as indicated by color) that shows a positive correlation between model's performance and the omission rate, especially in the poetry and numerical sequences domain. In other words, models actually perform *worse* when required to recall *fewer lines*. While this may initially appear counterintuitive, we hypothesize that a higher omission rate increases the likelihood of consecutive omissions occurring. Consequently, language models may be able generate text consistently until encountering a non-omitted element that disrupts their generation process. On the other hand, language models encounter greater difficulty with contexts containing fewer omissions, as numerous missing spans may interrupt the continuity between segments that must be generated.

| Models | Poetry | Numerical Sequences | GitHub PRs | Average |
|---|---|---|---|---|
| **Placeholder: None** (baseline) | | | | |
| Claude-3.7-Sonnet | 73.5 | 91.4 | 35.7 | 66.9 |
| GPT-4.1-mini | 30.2 | 45.0 | 31.3 | 35.5 |
| Llama-4-Maverick | 32.8 | 58.7 | 29.0 | 40.2 |
| **Placeholder: <missing line>** | | | | |
| Claude-3.7-Sonnet | $87.4_{(+18.9\%)}$ | $97.7_{(+6.9\%)}$ | $64.9_{(+81.8\%)}$ | $83.3_{(+24.5\%)}$ |
| GPT-4.1-mini | $50.4_{(+66.9\%)}$ | $59.2_{(+31.5\%)}$ | $46.8_{(+49.5\%)}$ | $52.1_{(+46.8\%)}$ |
| Llama-4-Maverick | $60.7_{(+85.0\%)}$ | $78.9_{(+34.4\%)}$ | $40.0_{(+37.9\%)}$ | $59.8_{(+48.9\%)}$ |
| **Placeholder: __________** | | | | |
| Claude-3.7-Sonnet | $85.5_{(+16.3\%)}$ | $97.2_{(+6.3\%)}$ | $61.3_{(+71.7\%)}$ | $81.3_{(+21.5\%)}$ |
| GPT-4.1-mini | $45.9_{(+52.0\%)}$ | $57.3_{(+27.3\%)}$ | $36.5_{(+16.6\%)}$ | $46.5_{(+31.2\%)}$ |
| Llama-4-Maverick | $53.9_{(+64.3\%)}$ | $73.0_{(+24.4\%)}$ | $35.2_{(+21.4\%)}$ | $54.0_{(+34.4\%)}$ |

Table 4: Micro F1-score of three LLMs evaluated on all three domains of Absence Bench using 2 different placeholders: "<missing line>" and "__________" (10 consecutive underlines), along with the percentage increase in F1-score compared to the none-placeholder baseline.

## 4.2 Marking omissions with placeholders

To assess whether the Transformer's attention mechanism is the reason for `AbsenceBench`'s difficulty, we explicitly mark omissions in the modified context using *placeholders*, special tokens that explicitly signal omitted segments. Specifically, we test across all domains (see Table 4 for full results) and consider two different placeholders: "<missing line>" and 10 consecutive underlines. Across all three domains, using "<missing line>" as the placeholder performs better, with an average boost in performance of 41.9%. Notably, Llama-4-Maverick achieves the highest improvement of 54.5% overall. The improved performance is comparable to that of o3-mini, yet o3-mini generates 9.1K extra thinking tokens on average.

Why does having a placeholder help so much? We hypothesize that `AbsenceBench` is so difficult for models because, when a segment of text is omitted, there is no position to attend to that corresponds directly to that omission. Transformer-based LLMs are incredibly good at finding the correct piece of information to attend to from a document, but what happens when the information is a conspicuous absence? What should the Transformer attend to? The fact that including a placeholder improves performance so drastically suggests that Transformer self-attention struggles to anchor on information gaps. This is a key area for future architecture and inference-time research, as omissions are common—from missing paperwork to comments deliberately left out of a recommendation letter.

## 5 Discussion

Our experiments highlight a failure of cutting-edge models to consistently succeed at the simple task of detecting absences, despite very strong performance on existing benchmarks. This suggests that popular evaluations do not effectively cover the capability of absence recognition, although it is fundamental for applications such as LLM-as-a-judge in which models must recognize which rubric elements are not covered. Our results are especially surprising given the simplicity of our evaluation: we only ask about surface-form absence (rather than absence on a higher semantic level), and directly provide models with both the original and modified documents for correct comparison. Indeed, our tasks could be solved by a simple program in linear time, yet deployed LLMs with hundreds of billions of parameters consistently struggle on them.

We propose an initial hypothesis explaining this behavior: identifying presence is simpler than absence with the attention mechanisms underlying Transformers [Vaswani et al., 2017]. Information included in a document can be directly attended to, while the absence of information cannot. We provide initial evidence of this hypothesis in §4.2. Rather than simply deleting elements from the original document, we include an absence "placeholder" here (e.g. "<missing line>") to provide a span of text for the transformer to attend to, indicating absence. This causes consistent, significant improvements for models, suggesting that the lack of a sequence to attend to is at least one aspect of this problem.

One important question raised by our work is how to make models more effective at handling and identifying absence. The strong performance of Gemini-2.5-flash in 2 of our 3 domains suggests that inference-time compute and reasoning mechanisms may help, although this may not hold in general. However there is a cost: we find that Gemini uses an extremely large number of reasoning tokens in many cases (Figure 3), in many cases exceeding the length of the original document by an order of magnitude. This could be the result of naive solutions, such as regenerating the full document multiple times. While this would work for the simple versions of absence studied here, it would likely fail on more complex notions of absence that look beyond missing surface forms.

We suggest a few directions where future work can go. Reasoning may improve absence detection, but must be studied on more complex, *semantic* notions of absence to be sufficiently validated—`AbsenceBench` is merely a necessary bar. New architectures, with mechanisms significantly different from attention, may be required to address the deeper problem. Such work should be supported by a more mechanistic understanding of why existing models often fail on these tasks, and how attention is connected to this question. We hope that `AbsenceBench` will provide a natural playground for such mechanistic studies. This will also be key in guiding how existing models are used–if they fail at these simple notions of absence, it is not clear that models will be trustworthy for more complex tasks that require this capability, such as model-as-a-judge evaluations using rubrics.

Regardless of how this problem might be solved, our work supports the notion that LLMs show *novel* intelligence, not well explained by analogy to humans. LLMs fail at this simple notion of absence while performing at superhuman levels on tasks that are quite similar (e.g. Needle-in-a-haystack) and many tasks that seem much more complex and challenging (e.g. math problem solving). The "shape" of LLM intelligence is simply not that similar to humans. While improving LLM performance on absence should be one goal, this also poses an opportunity to consider: how might we map the axes of variation in LLM intelligence?

Our findings suggest that models are not effective at understanding the conspicuous absence of information, an often overlooked yet foundational component of comprehension. Standard benchmarks overwhelmingly focus on whether models can identify and reason about what is present, which does not appear to tightly correlate with a model's ability to identify absent information. However use-cases such as LLM-as-a-judge, and tasks such as grading, legal reasoning, and misinformation detection, often hinge on recognizing what is missing. The gap in performance between NIAH and `AbsenceBench` underscores how misleading current evaluations might be if they ignore absence. Evaluators and developers should thus augment rubric-based assessments with systematic tests for absence sensitivity, both at surface and semantic levels. More broadly, absence may be a useful conceptual lens through which to understand model failure. Rather than treating hallucinations, misjudgments, or oversights as separate phenomena, they may all be related to a common limitation: models' weak grasp of what is not there. As a result, better understanding and diagnosing absence failure may reveal general-purpose principles for more robust and trustworthy LLM behavior.

## 6 Related Work

**Long-context Language Models** The context window of language models has significantly increased due to recent efforts in training long-context language models with optimized data mixture [Grattafiori et al., 2024, Gao et al., 2025, AI et al., 2025], exploring novel model architectures [Gu and Dao, 2024, Dao and Gu, 2024, Fu et al., 2023, Peng et al., 2023, Bertsch et al., 2023], developing encoding mechanisms [Su et al., 2023, Yen et al., 2024], attention algorithms that reduce memory footprints [Dao et al., 2022, Dao, 2023, Liu et al., 2023] as well as length extrapolation techniques [Press et al., 2022, Sun et al., 2022, Chen et al., 2023].

**Long-context Benchmarks** Many existing benchmarks evaluate language models' long-context abilities along multiple dimensions. ZeroSCROLLS [Shaham et al., 2023] focus on long-context information gathering tasks. Long-bench [Bai et al., 2024] introduces a bilingual benchmark and studies context-length variations. L-Eval [An et al., 2023] aim for a more diverse and high-quality collection of datasets. Infinite-Bench [Zhang et al., 2024] extend the context-length to over 100K tokens. HELMET [Yen et al., 2025] presents a holistic evaluation beyond synthetic data. Other previous work focus on specific tasks, such as retrieval-augmented generation [Lee et al., 2024], question answering [Kočiský et al., 2017, Wang et al., 2025], in-context learning [Agarwal et al., 2024, Anil et al., 2024, Xu et al., 2024], coreference resolution [Vodrahalli et al., 2024], and document

summarization[Chang et al., 2024, Kim et al., 2024]. Compared to those canonical tasks designed for benchmarking long-context understanding abilities, `AbsenceBench` features the task of handling long contexts at a finer granularity. Similar to the NIAH test [Kamradt, 2023, Yen et al., 2025] that evaluates the ability of LLMs to locate and retrieve crucial pieces of information, we formulate `AbsenceBench` into a much more straightforward task of identifying omissions in context, which turns out to be more challenging than identify the presence of information.

## 7 Limitations

While `AbsenceBench` exposes fundamental challenges in how current LLMs process omitted content, we note several limitations.

**Surface-form omission only.** We chose to keep the tasks exceedingly simple; even though our benchmark is solvable by simple heuristics models fail to use these heuristics. Our benchmark exclusively evaluates surface-form omission, where the removed content is a simple deletion. This simplification was intentional and helps make the evaluation unambiguous and trivially automatic. However, it does not capture more subtle or semantic notions of absence—such as omitted reasoning steps or missing evidence—which are critical in many real-world applications like rubric-based grading or code review.

**Structured task format.** `AbsenceBench` presents models with both the original and modified documents side-by-side and asks for the omitted elements directly. This setup may not reflect naturalistic settings where users do not provide such explicit comparisons. As a result, our evaluation may overestimate how well models would perform under more realistic situations with omitted information. Still, given the poor performance on `AbsenceBench`, current models are clearly not performant. However, if a new generation of models solve `AbsenceBench`, that will not be sufficient evidence to say that models can handle omission elegantly.

**Evaluation scope.** Our benchmark covers three domains (poetry, numerical sequences, and GitHub pull requests) with medium-length contexts (5K tokens). While this breadth offers early insight into the difficulty of absence detection, our findings may not generalize to other modalities (e.g., vision, audio), domains (e.g., legal, medical), or longer contexts. However, we expect these different settings to be more difficult, not less.

**No prompt or instruction tuning.** We only explored two different prompts and did not explore prompt tuning or in-context examples, due to the overwhelming cost of running long-context models, especially the most performant ones that make use of inference-time compute. Note that many inference-time models consistently use more the three times more "thinking tokens" than the original length of the document they are scanning for omissions. Thus, it remains an open question whether more elaborate prompting strategies, could significantly improve model performance, something we hope to examine in future work.

**Statistical significance.** Finally, we did not compute error bars or significance testing for our evaluation across runs due to API cost constraints. Although our findings are consistent across tasks and models, additional replication would help quantify variability and confirm robustness. We especially would have liked to run a document perturbation study, to see whether small variations in the document or prompt affect performance.

## 8 Conclusion

We introduce `AbsenceBench`, a benchmark that tests LLMs' ability to detect omitted information—an ability distinct from the well-studied task of recalling present content. Despite the benchmark's simplicity, models perform poorly, with surprising trends: fewer omissions make the task harder, and inference-time compute models often generate three times as many "thinking tokens" than the document length itself. Explicitly inserting placeholders where content is missing substantially improves performance, supporting the hypothesis that attention struggles to represent absence. These results reveal a core limitation in current models and motivate future work on absence-aware architectures and evaluation.

# 9 Acknowledgement

We thank OpenAI for API credits granted via their Researcher Access Program. We thank Aswathy Ajith, Todd Nief, and Chenghao Yang for their thoughtful feedback and suggestions.

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

# A Prompt Templates

We show the detailed prompt templates for language model evaluation across each domain included in `AbsenceBench` in Table 6 that are used to obtain the evaluation results in Table 3. Additionally, Table 7 presents prompt templates with adapted instructions used for assessing the NIAH test under the poetry and GitHub PRs domains.

---

**Poetry** (*default*)

---

**System Prompt**
```
You are helping a student practice memorizing poems.  The student will
recite a poem, but they may have missed some lines.  Your task is to
identify exactly which lines are missing from their recitation.
List only the missing lines, nothing else.
```
**User Message**
```
Here is the complete original poem:
```
{**original poem**}
```
Now, here is my recitation which may be missing some lines:
```
{**modified poem**}
```
What lines did I miss?  Please list only the missing lines, nothing else.
```

---

**Numerical Sequences** (*default*)

---

**System Prompt**
```
You are helping a student practice reciting sequences.  The student will
recite a sequence, but they may have missed some numbers.  Your task is to
identify exactly which numbers are missing from their recitation.
List only the missing numbers, nothing else
```
**User Message**
```
Here is a sequence of numbers:
```
{**original sequence**}
```
Now, here is my recitation of the sequence which may be missing some numbers:
```
{**modified sequence**}
```
What numbers did I miss?  Please list only the missing numbers, nothing else.
```

---

**GitHub PRs** (*default*)

---

**System Prompt**
```
You are helping a software developer determine if their merge of a pull
request was successful.  The developer had to edit the commit history
and just wants to make sure that they have not changed what will be merged.
They will list the changed lines.  Your job is to figure out if they have
missed any insertions or deletions from the original merge.  Only pay
attention to the insertions and deletions (ignore the context of the diff).
```
**User Message**
```
Here is the complete original diff:
```
{**original diff**}
```
And here is the merge diff after the developer fixed the commit history:
```
{**modified diff**}
```
What changed lines (insertions or deletions) present in the original diff
are missing in the merge diff (if any)?
List only the missing changed lines, nothing else.
```

---

Table 5: Detailed prompt templates used for evaluating language models under each domain of `AbsenceBench`

**Poetry** (*post-instruction*)

**System Prompt**
```
You are helping a student practice memorizing poems.  The student will
recite a poem, but they may have missed some lines.  Your task is to
identify exactly which lines are missing from their recitation.
List only the missing lines, nothing else.
```
**User Message**
{**original poem**}

{**modified poem**}
```
The original poem is followed by my recitation of the poem which may be
missing some lines.
What lines did I miss?  Please list only the missing lines, nothing else.
```

**Numerical Sequences** (*post-instruction*)

**System Prompt**
```
You are helping a student practice reciting sequences.  The student will
recite a sequence, but they may have missed some numbers.  Your task is to
identify exactly which numbers are missing from their recitation.
List only the missing numbers, nothing else
```
**User Message**
{**original sequence**}

{**modified sequence**}
```
The original sequence is followed by my recitation of the sequence which
may be missing some numbers.
What numbers did I miss?  Please list only the missing numbers, nothing else.
```

**GitHub PRs** (*post-instruction*)

**System Prompt**
```
You are helping a software developer determine if their merge of a pull
request was successful.  The developer had to edit the commit history
and just wants to make sure that they have not changed what will be merged.
They will list the changed lines.  Your job is to figure out if they have
missed any insertions or deletions from the original merge.  Only pay
attention to the insertions and deletions (ignore the context of the diff).
```
**User Message**
{**original diff**}

{**modified diff**}
```
The original diff is followed by a merge diff after the developer fixed
the commit history
What changed lines (insertions or deletions) present in the original diff
are missing in the merge diff (if any)?
List only the missing changed lines, nothing else.
```

Table 6: Perturbed prompt templates where the task instructions are positioned after the original and modified documents.

**Poetry** (NIAH)

---

**System Prompt**
```
You are helping a student practice memorizing poems.  The student will
recite a poem, but they may have added some random lines that related
to Harry Potter characters.  Your task is to identify exactly which lines
are not in the original poem.  List only the missing lines, nothing else.
```
**User Message**
```
Here is the complete original poem:
```
{original poem}
```
Now, here is my recitation with some extra lines that is related to
Harry Potter novel series:
```
{modified poem}
```
What lines did I miss?  Please list only the extra lines, nothing else.
```

---

**GitHub PRs** (NIAH)

---

**System Prompt**
```
You are helping a software developer determine if their merge of a pull
request was successful.  The developer had to edit the commit history
and accidently added some random lines related to Harry Potter characters.
They will list the changed lines.  Your job is to figure out if they have
added any insertions from the original merge.  Only pay attention to the
insertions.
```
**User Message**
```
Here is the complete original diff:
```
{original diff}
```
And here is the merge diff after the developer fixed the commit history:
```
{modified diff}
```
What changed lines (insertions or deletions) present in the original diff
are missing in the merge diff (if any)?
List only the missing changed lines, nothing else.
```

---

Table 7: Modified prompt templates used for evaluating language models under the NIAH test setting within the Poetry and GitHub PRs domains.

# B    Inference Details

We evaluate a total of 14 LLMs on `AbsenceBench`. All model inferences are obtained through API requests. We show the detailed information of each model's maximum context length, API provider and reference in Table 8. We fix the generation temperature at the default value of 1.0 across all API providers. For models with inference-time compute, we adopt the default thinking budget set by these API providers: Google Gemini (2.5-flash; dynamic with an upper bound of 24576 tokens), OpenAI (o3-mini; medium reasoning effort), TogetherAI (Qwen3 models and Deepseek R1). We also manually set thinking budget for Claude-3.7-Sonnet to be 10K tokens and Grok-3-mini-beta to be "low reasoning effort" due to cost considerations.

| Models | Context Length | API Provider | API Reference |
|---|---|---|---|
| Gemini-2.5-flash [Google, 2025] | 1M | Google | models/gemini-2.5-flash-preview-04-17 |
| Claude-3.7-Sonnet [Anthropic, 2024] | 200K | Anthropic | claude-3-7-sonnet-latest |
| o3-mini [OpenAI, 2025] | 200K | OpenAI | o3-mini |
| GPT-4.1 [OpenAI et al., 2024b] | 1M | OpenAI | gpt-4.1 |
| GPT-4.1-mini [OpenAI et al., 2024b] | 1M | OpenAI | gpt-4.1-mini |
| GPT-4o [OpenAI et al., 2024a] | 128K | OpenAI | gpt-4o |
| Grok-3-mini-Beta [xAI, 2025] | 131K | xAI | grok-3-mini-beta |
| Llama-4-Maverick [Meta, 2025] | 1M | TogetherAI | meta-llama/Llama-4-Maverick-17B-128E-Instruct-FP8 |
| Llama-3.3-70B-Instruct [Grattafiori et al., 2024] | 131K | TogetherAI | meta-llama/Llama-3.3-70B-Instruct-Turbo |
| Llama-3.1-8B-Instruct [Grattafiori et al., 2024] | 128K | TogetherAI | meta-llama/Meta-Llama-3.1-8B-Instruct-Turbo |
| Qwen3-235B [Qwen, 2025a] | 41K | TogetherAI | Qwen/Qwen3-235B-A22B-fp8-tput |
| Qwen2.5-72B-Instruct [Yang et al., 2024] | 32K | TogetherAI | Qwen/Qwen2.5-72B-Instruct-Turbo |
| QwQ-32B [Qwen, 2025b] | 32K | TogetherAI | Qwen/QwQ-32B |
| DeepSeek-R1 [DeepSeek-AI et al., 2025] | 128K | TogetherAI | deepseek-ai/DeepSeek-R1 |
| Mixtral-8x7B-Instruct [Jiang et al., 2024] | 32K | TogetherAI | mistralai/Mixtral-8x7B-Instruct-v0.1 |

Table 8: Detailed information of models evaluated on `AbsenceBench`

## C  Alternative Prompt Template

Alongside the default prompt template (see Table 2), we curate a perturbed template where the task instructions are positioned after the original and modified documents. We refer to this perturbed prompt template as "post-instruction". Table 9 shows that the performance of three LLMs prompted with both prompt templates. All three models exhibit reduced performance with the post-instruction template under both the poetry and numerical sequences domains, suggesting that LLMs more effectively detect omissions when instructions are presented beforehand. Meanwhile, LLMs remain robust in the GitHub PRs domain. All detailed prompt templates are included in Appendix A.

| Models | Poetry | | Numerical Sequences | | GitHub PRs | |
|---|---|---|---|---|---|---|
| | *default* | *post-instruction* | *default* | *post-instruction* | *default* | *post-instruction* |
| Claude-3.7-Sonnet | 78.8 | 66.1 | 94.3 | 93.7 | 35.1 | 34.7 |
| GPT-4.1-mini | 45.4 | 39.2 | 54.2 | 42.5 | 32.9 | 35.4 |
| Llama-4-Maverick | 48.7 | 36.6 | 71.5 | 63.6 | 29.6 | 30.5 |

Table 9: Micro F1-score (%) of three LLMs evaluated on the `AbsenceBench` with two different prompt templates. All three models exhibit reduced performance with the *post-instruction* template in the poetry and numerical sequences domains, but remain robust in the GitHub PRs domain.

## D  Dataset Construction Parameters

Our main considerations for the selection of dataset construction parameters were cost, balancing difficulty, and ensuring a valid evaluation, all described below:

- Cost and Practicality: our poetry source data includes poems of widely varying lengths, with some extending beyond 5000 lines (145 out of 1187). Without truncation, the benchmark would be technically not applicable for LLMs with a shorter context-length limit.
- Utility and Difficulty Balance: according to our perturbation studies in Section 4, further extending the context length would add to the difficulty of AbsenceBench, but is not necessary since the benchmark is already highly challenging to leading LLMs at these medium context lengths (see Table 2). We believe balancing the utility and difficulty of the benchmark is important, leading us to the current medium-context.
- Validity: the selected length range ensures that instances are neither trivially short nor excessively long, thus avoiding scenarios where context length would become the dominant confounder in model accuracy. On the other hand, the benchmark only focuses on surface-level omissions. Truncating the instances would not bring any semantic concern to the validity of the benchmark.

## E  Comparison to the NIAH test

We extend the analysis of Section 3.2 on the comparison between the task design of `AbsenceBench` and the NIAH test. In table 10, we show 3 LLMs achieve near perfect results in the NIAH test under the poetry domain, as well as substantially improved results within the GitHub PRs domain.

| Models | Poetry | Poetry (NIAH) | GitHub PRs | GitHub PRs (NIAH) |
|---|---|---|---|---|
| Claude-3.7-Sonnet | 73.5 | 99.52 | 35.7 | 97.1 |
| GPT-4.1-mini | 30.2 | 99.5 | 31.3 | 92.1 |
| Llama-4-Maverick | 32.8 | 99.47 | 29.0 | 86.2 |

Table 10: Micro F1-score (%) of three LLMs evaluated on the `AbsenceBench` and the NIAH test setting under the Poetry and GitHub PRs domains.

# F  Additional Results

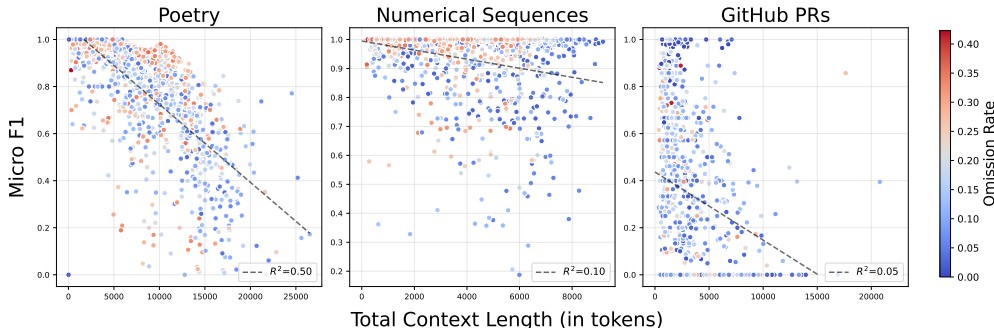

Figure 6: Micro-F1 score of **Claude-3.7-Sonnet** (y-axis) as a function of the total context length (x-axis) as well as the percentage of omission (color)

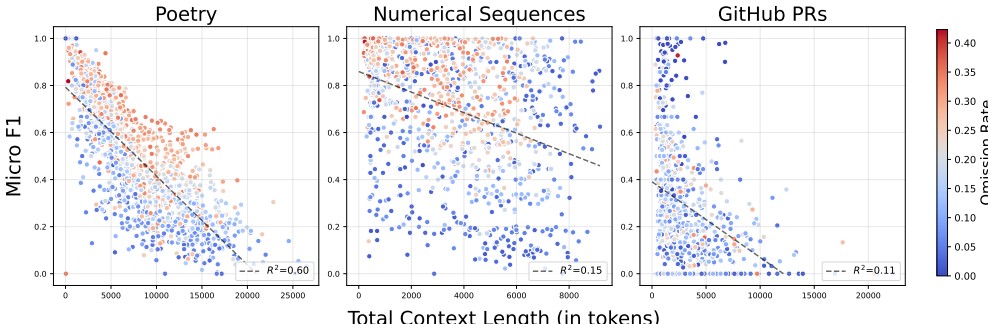

Figure 7: Micro-F1 score of **Llama-4-Maverick** (y-axis) as a function of the total context length (x-axis) as well as the percentage of omission (color)

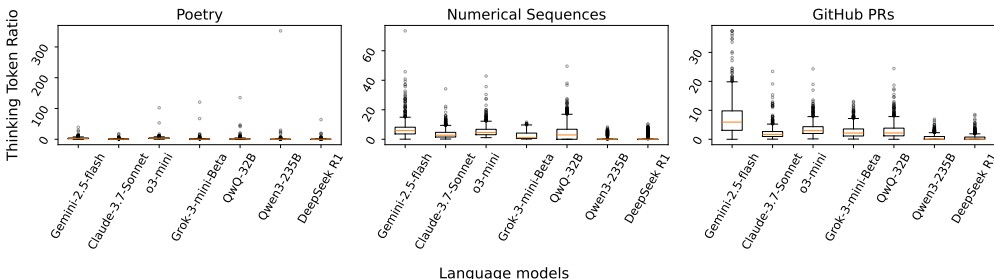

Figure 8: The thinking token ratio (number of generated thinking tokens divided by number of tokens in the original context) for four inference-time compute models under each domain.

