# OpenReview forum: "Absence Bench: Language Models Can’t See What’s Missing"
_NeurIPS.cc/2025/Datasets_and_Benchmarks_Track — NeurIPS 2025 Datasets and Benchmarks Track spotlight_

### Official Review · Reviewer_6op7 · 2025-06-08

**Rating:** 5
**Confidence:** 3

**Summary:**

The paper proposes AbsenceBench, a new benchmark designed to evaluate the abilities of LLMs in locating conspicuously missing information from long inputs. The paper demonstrates that even state-of-the-art models like Claude-3.7-Sonnet achieve only 52.4% F1-score on the proposed datasets with modest average context lengths of 5K tokens. The authors propose a hypothesis and conduct extensive experiments to illustrate that the Transformer attention mechanism struggles to handle “gaps” in the input, which is one of the reasons for failure.

**Dataset Code Accessibility:**

Yes

**Dataset Code Comments:**

The dataset has been made publicly available on Kaggle, and the code has been released on the Supplementary Material. I have reviewed the code and datasets and believe the paper is reproducible.

**Ethical Considerations:**

No, there are no or only very minor ethics concerns

**Final Justification:**

The paper focuses on an important problem that LLMS struggle to identify clearly omitted information. The paper provides a valuable dataset of 4K long-context data across numerical sequences, poetry, and GitHub pull requests, providing a long-context omission detection benchmark. The paper validates its claims through comprehensive experiments using 14 models. The dataset has been made publicly available on Kaggle, and the code has been released on the Supplementary Material. My concerns are addressed well. Overall, I would like to support this paper's acceptance.

**Limitations Weaknesses:**

1. The paper finds that state-of-the-art models with context length of over 32K only achieve a poor performance F1 score with an average context length of approximately 5K tokens. It remains to be verified whether this finding holds across LLMs with different context lengths (e.g., 8K, 16K, 32K tokens).
2. It is better that the authors further illustrate or validate the hypothesis using mathematical formulas to enhance its theoretical rigor.
3. All experiments are conducted via APIs. Whether these experiments can be extended to locally open-source models, such as Llama-3.1-8B-Instruct and Gemma-2-9b-it.

**Strengths Contributions:**

1. Important Problem: The paper focuses on an important problem that LLMS struggle to identify clearly omitted information. The authors validate the problem’s importance through extensive experiments.
2. Interesting hypothesis: The hypothesis presented in Section 4.2 is novel and interesting. Solid experiments in Appendix B effectively support this hypothesis.
3. Useful dataset: The paper provides a valuable dataset of 4K long-context data across numerical sequences, poetry, and GitHub pull requests, providing a long-context omission detection benchmark.
4. Extensive Experimental Evaluation: The paper validates its claims through comprehensive experiments using 14 models.

---

> ### Author Rebuttal · Authors · 2025-07-31
>
> We thank the reviewer for the thoughtful questions and comments! We are glad that the reviewer found our paper useful, interesting, and concrete. Here are our responses and clarifications to the individual questions:
>
> ---
> > W1: The paper finds that state-of-the-art models with context length of over 32K only achieve a poor performance F1 score with an average context length of approximately 5K tokens. It remains to be verified whether this finding holds across LLMs with different context lengths (e.g., 8K, 16K, 32K tokens).
>
> Thank you for raising this insightful question. We anticipate consistent findings for LLMs with a shorter context-length limit according to the findings in the RULER benchmark paper [1]. Their findings indicate that, in practice, most long-context language models demonstrate a shorter effective context length than the claimed ones. Since AbsenceBench has an average context length of 5K tokens, we intentionally selected LLMs with greater maximum context lengths to ensure these models could fully leverage their capabilities. To further support this, we have evaluated GPT-3.5-Turbo (16K context window) on AbsenceBench. Please note that we are only able to evaluate instances that fit within a 16K token context length, rather than the entire dataset. Here are the Micro-F1 scores on all domains:
>
> | Poetry|Numerical Sequences|GitHub PRs|Average|
> | :--------: | :-------: | :--------: | :--------: |
> | 6.1% | 23.7% | 20.2% | 16.7% |
>
> The performance of GPT-3.5-Turbo is worse than all other LLMs except for Mixtral-8x7B-Instruct, which suggests that models with smaller context-length may find AbsenceBench more challenging than long-context LLMs. We are not able to include this piece of result in the paper since the evaluation is not on the full dataset, and we will make sure to mention a minimum of 32K context length is required to evaluate on AbsenceBench in the next revision.
>
> ---
> >W2: It is better that the authors further illustrate or validate the hypothesis using mathematical formulas to enhance its theoretical rigor.
>
> Thank you for the constructive suggestion and we agree that mathematical formulas would provide theoretical support for our hypothesis and claims. We would like to clarify that our analysis in Section 4.2 is intended as a preliminary exploration rather than a comprehensive explanation for the claim. To provide more empirical support for our claim, we have extended the scale of our analysis by **ninefold** (all three tasks for Claude-3.7-Sonnet, GPT-4.1-mini, and Llama-4-Maverick). The analysis indicates that attention mechanisms might be a bottleneck for solving AbsenceBench.
>
> | Models | Poetry | Numerical Sequences | GitHub PRs | Average |
> |---------------------|-----------|---------------------|------------|-----------|
> | **Placeholder: None (baseline)** | | | | |
> | Claude-3.7-Sonnet | 73.5 | 91.4 | 35.7 | 66.9 |
> | GPT-4.1-mini | 30.2 | 45.0 | 31.3 | 35.5 |
> | Llama-4-Maverick | 32.8 | 58.7 | 29.0 | 40.2 |
> | **Placeholder: `<missing line>`** | | | | |
> | Claude-3.7-Sonnet | 87.4 (+18.9%) | 97.7 (+6.9%) | 64.9 (+81.8%) | 83.3 (+24.5%) |
> | GPT-4.1-mini | 50.4 (+66.9%) | 59.2 (+31.5%) | 46.8 (+49.5%) | 52.1 (+46.8%) |
> | Llama-4-Maverick | 60.7 (+85.0%) | 78.9 (+34.4%) | 46.8 (+61.4%) | 62.1 (+54.5%) |
> | **Placeholder: `__________`** | | | | |
> | Claude-3.7-Sonnet | 85.5 (+16.3%) | 97.2 (+6.3%) | 61.3 (+71.7%) | 81.3 (+21.5%) |
> | GPT-4.1-mini | 45.9 (+52.0%) | 57.3 (+27.3%) | 36.5 (+16.6%) | 46.5 (+31.2%) |
> | Llama-4-Maverick | 53.9 (+64.3%) | 73.0 (+24.4%) | 36.5 (+25.9%) | 54.5 (+35.5%) |
>
> As shown in the table, explicitly marking omissions in the modified context using placeholders significantly boost all three models’ performance by 41.9% on average. Based on the observation, we hypothesize that this improvement is due to the attention mechanism effectively anchoring to these placeholders.
> Understanding and theorizing about the reasoning abilities of LLMs—particularly why they encounter difficulties with seemingly simple tasks—would provide important insights. We see this as a promising direction for future research.
>
> ---
> >W3: All experiments are conducted via APIs. Whether these experiments can be extended to locally open-source models, such as Llama-3.1-8B-Instruct and Gemma-2-9b-it.
>
> We appreciate the suggestion to extend the experiment to locally open-source models. We chose to use APIs for our experiments primarily due to concerns about computational resources and efficiency. However, in our final code release, we will make sure to include a pipeline that enables evaluation on locally deployed LLMs.
>
> [1] Hsieh, Cheng-Ping, et al. "RULER: What's the Real Context Size of Your Long-Context Language Models?." arXiv preprint arXiv:2404.06654 (2024).

---

> > ### Comment · Reviewer_6op7 · 2025-08-02
> >
> > My concerns are addressed well. However, I believe that models like Llama-3.1-8B-Instruct and Gemma-2-9b-it do not cause high computational cost. Including such results from open-source LLMs can help us better evaluate the reliability and reproducibility of the results. Overall, I would like to support this paper's acceptance.

---

> > ### Author Response · Authors · 2025-08-04
> >
> > Thank you for your appreciation of our work! We’re glad that our additional clarifications helped resolve your concerns.
> >
> > We completely agree that incorporating results from open-source LLMs enhances benchmark validity. In fact, 7 out of 14 LLMs used in our evaluation are open-sourced. We will include the evaluation results of Llama-3.1-8B-Instruct in the next revision. However, models like Gemma-2-9b-it, despite their low usage costs, are constrained by a maximum context length of 8192 tokens. As noted in our response to W2 in the rebuttal, this limitation prevents us from evaluating them on the full dataset and would compromise the comparability of the results. However, if there are any models with at least 32k context length that you would like us to include in the final manuscript, we would be more than happy to add them!

---

> > > ### Comment · Reviewer_6op7 · 2025-08-04
> > >
> > > Thank you for your detailed response. My concerns have been adequately addressed.  I would like to increase my score.

---

> > > > ### Author Response · Authors · 2025-08-04
> > > >
> > > > We truly appreciate your recognition of this paper and we are glad our additional explanations have addressed your concerns. We will ensure that all revisions promised in our rebuttal are included in the next version.

---

### Official Review · Reviewer_hDBY · 2025-06-24

**Rating:** 4
**Confidence:** 3

**Summary:**

This work first provides a novel benchmark demonstrating the inability of large language models (LLMs) to detect missing tokens. Secondly, it analyzes the underlying reason for this limitation, attributing it to the absence of attended keys.

**Dataset Code Accessibility:**

Partly

**Dataset Code Comments:**

The complete Python scripts to make use of the files have not been released yet.

**Ethical Comments:**

This work is totally technical.

**Ethical Considerations:**

No, there are no or only very minor ethics concerns

**Final Justification:**

The significance of this work is clarified by 'LLM as a judge'-style applications. Therefore, I raise the score from borderline reject to borderline accept and encourage the authors to emphasize this significance throught the whole paper (especially in introduction).

**Limitations Weaknesses:**

1. The significance of the discussed inability of LLMs is not adequately clarified: further demonstrations (better if experimantally) are necessary. Besides, the introduction lacks a discussion on its importance, relegating this to the discussion section, which may leave readers uncertain about its potential implications until the end of the paper.
2. The reasons behind LLMs' success in some experiments (without placeholders) are unclear. It would be beneficial to investigate the strategies these models employ in successful cases, necessitating case studies for clarity.
3. The statement that "models fail for completely different reasons than humans do" requires human baselines for comparison.
4. The section on long-context language models in the related work is poorly written, lacking a discussion on the relationship between these works and the Absence Benchmark.
5. The code is currently unreproducible.

**Strengths Contributions:**

1. The task is simple yet effectively reveals a potentially fundamental limitation of large language models (LLMs).
2. The benchmark is robust, and the main results are well-supported.
3. The presentation is clear and well-written.

---

> ### Author Rebuttal · Authors · 2025-07-31
>
> We thank the reviewer for the thoughtful questions and comments! Here are our responses to your several concerns.
>
> ---
> > W1: The significance of the discussed inability of LLMs is not adequately clarified: further demonstrations (better if experimantally) are necessary.
>
> Thank you for recognizing the potential implications of the benchmark. We have completed several additional experiments to demonstrate our experiment results are robust:
> We expanded the analysis in Section 4 **ninefold** by evaluating three LLMs (Claude-3.7-Sonnet, GPT-4.1-mini, and Llama-4-Maverick) across all three domains. The experimental results show that our observations about the influence of context length and omission rate remain robust across various settings.
> We added a perturbation study on the choice of prompt template.
> We included a regression analysis on the influence of context-length to support our perturbation study.
> Here we present additional experiment results to Section 4.2 with consistent observations:
>
> | Models | Poetry | Numerical Sequences | GitHub PRs | Average |
> |---------------------|-----------|---------------------|------------|-----------|
> | **Placeholder: None (baseline)** | | | | |
> | Claude-3.7-Sonnet | 73.5 | 91.4 | 35.7 | 66.9 |
> | GPT-4.1-mini | 30.2 | 45.0 | 31.3 | 35.5 |
> | Llama-4-Maverick | 32.8 | 58.7 | 29.0 | 40.2 |
> | **Placeholder: `<missing line>`** | | | | |
> | Claude-3.7-Sonnet | 87.4 (+18.9%) | 97.7 (+6.9%) | 64.9 (+81.8%) | 83.3 (+24.5%) |
> | GPT-4.1-mini | 50.4 (+66.9%) | 59.2 (+31.5%) | 46.8 (+49.5%) | 52.1 (+46.8%) |
> | Llama-4-Maverick | 60.7 (+85.0%) | 78.9 (+34.4%) | 46.8 (+61.4%) | 62.1 (+54.5%) |
> | **Placeholder: `__________`** | | | | |
> | Claude-3.7-Sonnet | 85.5 (+16.3%) | 97.2 (+6.3%) | 61.3 (+71.7%) | 81.3 (+21.5%) |
> | GPT-4.1-mini | 45.9 (+52.0%) | 57.3 (+27.3%) | 36.5 (+16.6%) | 46.5 (+31.2%) |
> | Llama-4-Maverick | 53.9 (+64.3%) | 73.0 (+24.4%) | 36.5 (+25.9%) | 54.5 (+35.5%) |
>
> We promise to include other experimental results as well in the next revision.
>
> > Besides, the introduction lacks a discussion on its importance, relegating this to the discussion section, which may leave readers uncertain about its potential implications until the end of the paper.
>
> We apologize for not highlighting the importance and broader implications of our work more clearly in the introduction. We recognize the importance of framing the introduction with a discussion of the broader implications and will include a paragraph that briefly summarizes the discussion section.
>
> ---
> > W2: The reasons behind LLMs' success in some experiments (without placeholders) are unclear. It would be beneficial to investigate the strategies these models employ in successful cases, necessitating case studies for clarity.
>
> Thank you for the insightful suggestion. We fully agree that more in-depth case studies would be beneficial for understanding the success/failure modes of LLMs. However, simply analyzing responses at the surface or lexical level may not provide sufficient insight. Here are two examples when QwQ-32B succeeded and failed to identified omitted numbers from a sequence
>
> > Success: Original 3585 is present. Then **3587**, but recited skips to 3589. So **3587** is missing.
>
> > Fail: Wait, looking at the recited list: after 3551, the next is 3553, then **3555**, 3557.
>
> The model appears to be reasoning about omissions but could hallucinate as they fail to do so. We believe deeper analysis on reasoning patterns would be helpful to analyze success/failure modes on AbsenceBench, which we hope can be addressed in future work.
>
> ---
> >W3: The statement that "models fail for completely different reasons than humans do" requires human baselines for comparison.
>
> We apologize and acknowledge that this statement sounds like an overclaim without human baselines as a comparison. To clarify, we took a natural and intuitive guess of human behavior: going line by line through two documents. We also assumed the human baseline to be near perfect due to the simplicity of the given task. We will make sure to revise the claim in our next revision.
>
> ---
> >W4: The section on long-context language models in the related work is poorly written, lacking a discussion on the relationship between these works and the Absence Benchmark.
>
> We appreciate the reviewer for pointing out this issue. AbsenceBench shares a similar design with the Needle in a Haystack task, which has been recognized as a challenging task for evaluating long-context language models. For our experimental setup, we specifically chose long-context LLMs with at least a 32K token context window to ensure comparability with other long-context benchmarks. Notably, we find that AbsenceBench remains highly challenging even at much shorter context lengths. We will ensure these connections are explicitly discussed in the Related Work section in our next revision.
>
> ---
> > W5: The code is currently unreproducible.
>
> We will make sure to open-source the benchmark and fully release our code for dataset construction, evaluation, and analysis.

---

> > ### Comment · Reviewer_hDBY · 2025-08-02
> >
> > My primary concern remains unaddressed: the significance of the limitations of LLMs must be clearly demonstrated through experimental evidence. Without this, assessing the importance of this work becomes challenging, as many limitations of LLMs exist, but only a few are of academic interest. It’s important to note that the robustness of the results does not necessarily enhance their significance.

---

> > > ### Author Response · Authors · 2025-08-03
> > >
> > > Thank you for emphasizing this question—we agree it is an important aspect of this work to address. We would like to highlight our experiment section (Section 3; Table 2) where we evaluated a total of 14 frequently used LLMs on AbsenceBench. Our task design is closely aligned with real-world use cases, and uses LLM-as-a-judge style prompts (see Appendix A in the supplementary materials). Our results directly reflect the fact that LLMs find tasks that require identifying missing information challenging.
> > >
> > > Here are a few examples of real-world use cases where our work is closely aligned with, and can be practically useful and potentially aligned with academic interest:
> > > - LLM-as-a-judge: LLMs are likely to fail at grading students’ recitations if they cannot effectively identify omissions.
> > > - Document and code reviewing/auditing: automated review of pull requests, legal documents, medical records, or contracts often requires ensuring nothing important is missing, but LLMs may not be able to flag them. For instance, if a user asks an LLM to compare an original and updated privacy policy, even SoTA LLMs may not report missing lines in the update policy.
> > > - Safety and security: LLMs may fail to recognize important pieces of safety instructions that are missing from a manual, or in a prompt that is used for training another LLM. We also included some related discussions in Section 5 (Line 242).

---

> > > > ### Comment · Reviewer_hDBY · 2025-08-03
> > > >
> > > > The significance of this work is clarified by 'LLM as a judge'-style applications. Therefore, I raise the score from borderline reject to borderline accept and encourage the authors to emphasize this significance throught the whole paper (especially in introduction).

---

> > > > > ### Author Response · Authors · 2025-08-03
> > > > >
> > > > > Thank you for acknowledging the importance of our work. We will ensure to highlight its significance in the introduction and to incorporate the revisions promised in our rebuttal. We truly appreciate your feedback and are pleased that our additional explanations have addressed your concerns.

---

### Official Review · Reviewer_shFv · 2025-07-03

**Rating:** 5
**Confidence:** 4

**Summary:**

The paper proposes a new benchmark for long context processing capabilities of LLMs. Previous long context benchmarks focus on finding information that may be present at one or many places in a long text (needle in the haystack). The paper proposes an opposite task, asking the model to retrieve information that was deleted. The task is quite simple and intuitive and any human should be able to perform it. Surprisingly, frontier LLMs perform quite poorly. The paper posits that the reason is the design of the attention mechanism which is designed to attend to presence of information, and not absence of it.

**Additional Feedback:**

1. Line 68: “arking” -> “”marking
2. Line 84: The paper says that the poems are truncated to ensure a uniform distribution. However, Figure 2 does not really look uniform. Would be good to add some detail here.
3. It would be great to analyze the failure modes. For instance, some simple lexical analysis describing the responses when the model fails to recall the deleted sequence.
4. Not sure if I understood Figure 4 properly. The text in line 170 says that the performance starts to decline after 15K tokens. At that point, the micro F1 is already quite close to 0 for 2/3 datasets and is quite uniformly distributed for the numerical dataset.
5. Would be nice to have some explanation on why these three domains were chosen. The insight on removal causing issues for attention mechanism seems quite general. So some discussion on how the observation would generalize to other domains would be great.

**Dataset Code Accessibility:**

Partly

**Dataset Code Comments:**

I could not find links to the raw GitHub PR and Poetry datasets -- only the processing code is provided. Would be great if the authors could comment on it.

**Ethical Considerations:**

No, there are no or only very minor ethics concerns

**Final Justification:**

The benchmark makes a solid insight into a specific shortcoming of LLMs. The rebuttal addressed most of my concerns, as noted in my response. The analysis around the attention mechanism could be strengthened but is still appropriate for the D&B track.

**Limitations Weaknesses:**

1. Sometimes the writing is a bit hurried and important choices aren’t explained well. For examples, the omission probability discussion in Figure 2 didn’t quite make sense. A heat map legend would have made it easier to read the exact values from the plot. Similarly, what was the thinking budget assigned to the models? What temperature setting was used? Also, what is meant by “various factors” in line 163. Just spelling out these factors and the hypothesis that the paper was trying to test would be great.
2. Parameter choices are a bit ad hoc. For instance, in line 85, it is not described why the range 10 to 1000 was chosen. In line 97, it is not clear why the numbers 10 and 200 were chosen. I can understand that the paper wants to limit the analysis to medium context length (5K tokens), but it’s fine if the benchmark released has a more diverse range of lengths. More diversity will only increase the utility of the benchmark for others.
3. It would really help the reader if the formulas for metrics, not matter how simples these are, were included in the paper. That would shed light on details like “what tokenizer is used before applying F1” and “is the exact match case insensitive?”. Also, it is not clear to this reviewer why a recall metric would lead to false positives. Aren’t we just checking the recall of the generation against the deleted sequence?
4. The claim on line 172 needs more justification. Did a similar performance decline happen when using the exact match score? Just wanted to make sure that we are somehow not capturing an artifact of the F1 score itself. At larger lengths, I would expect higher occurrence of common tokens like “is”, “a” and “the”. Since F1 operates over sets, it might artificially deflate the presence of these common tokens.
5. Some discussion into why the benchmark relates to real world tasks would be great. While the omission task is one that we could expect a human to perform quite well, if they failed at this task, would it lead them to perform poorly on some real world tasks? Examples of such tasks would be great.
6. Not sure the claim regarding attention (starting line 182) is fully supported by the evidence. If the attention cannot process absent tokens, why does test time compute improve the performance. In any case, since the prompt contains both the original text and the truncated text, the information that the attention mechanism needs is still presented in the original text. So it is likely that the reason for drop in performance might be something completely different. Further explanation from authors here would be greatly helpful.

**Strengths Contributions:**

1. Overall, the benchmark design is quite simple and intuitive.
2. The main finding that even with very conspicuous deletion of information, the LLMs cannot identify absences is very compelling.
3. The inference time compute analysis results are intriguing, though not sure the fundamental reason behind them is clear.
4. The attention mechanism insight and the analysis in Section 4.2 is interesting and might spur follow on work on model architecture.

---

> ### Author Rebuttal · Authors · 2025-07-31
>
> We thank the reviewer for the thoughtful questions and comments! We are glad that the reviewer found our paper interesting and insightful. Below are our responses and clarifications to the individual questions:
>
> W1: Thank you for these careful observations and attention to clarity, and we apologize for the confusion caused by the current version. We will make the following updates in the next revision:
> - Figure 2 and omission probability discussion. Figure 2 was meant to serve as a histogram displaying the distribution of document lengths across different tasks. To avoid confusion, we will remove the color labeling and omission probability discussion here and direct readers to the ablation study in Section 4.1 for omission probability discussion.
> - Experiment details. We acknowledge these hyperparameters are crucial to reproduce the results of the paper, and will include them in the next revision and the code release.
>     - Thinking budget: we take the default thinking budget set by these API providers: Google Gemini (2.5-flash; dynamic with an upper bound of 24576 tokens), OpenAI (o3-mini; medium reasoning effort), TogetherAI (Qwen3 models and Deepseek R1). We also manually set thinking budget for Claude-3.7-Sonnet to be 10K tokens and Grok-3-mini-beta to be “low reasoning effort” due to cost considerations
>     - Temperature setting: we take the default temperature setting shared by all API providers (1.0)
> - Various factors. Thank you for pointing out the ambiguity. By “various factors,” we mean context length and omission rate. Both of these are analyzed in detail in Section 4, where we test for their individual impacts on model performance. We will make sure the terms are specified in our next revision. **Additionally**, we have performed an ablation study on the choice of prompt template, which we will include in Section 4.
>
> W2: We agree that the choice of data construction parameters could be better justified. Our main motivations were cost, balancing difficulty, and ensuring a valid evaluation, all described below:
> - Cost and Practicality: our poetry source data includes poems of widely varying lengths, with some extending beyond 5000 lines (145 out of 1187). Without truncation, the benchmark would be technically not applicable for LLMs with a shorter context-length limit.
> - Utility and Difficulty Balance: according to our perturbation studies in Section 4, further extending the context length would add to the difficulty of AbsenceBench, but is not necessary since the benchmark is already highly challenging to leading LLMs at these medium context lengths (see Table 2). We believe balancing the utility and difficulty of the benchmark is important, leading us to the current medium-context.
> - Validity: the selected length range ensures that instances are neither trivially short nor excessively long, thus avoiding scenarios where context length would become the dominant confounder in model accuracy. On the other hand, the benchmark only focuses on surface-level omissions. Truncating the instances would not bring any semantic concern to the validity of the benchmark.
>
> W3: We are sorry about the confusion. To clarify, we use F1-score to evaluate the generated response on an element level (e.g., a line of a poem), which are already decoded by the default tokenizer employed by each model/API provider. The reason we did not use Recall as the evaluation metric is that it would make it a trivial task if LLMs simply copy-paste the original context and achieve perfect performances. Instead we apply F1-scores that account for false-positives when LLMs generate an element that is not omitted. We use case-insensitive exact match on the element level to detect whether the element appears in the generated response, to which by then we can apply the F1-score. We will make sure to include these details in the next revision.
>
> W4: Thank you for pointing out this ambiguity. The evaluation metric of AbsenceBench is F1-score, while we used exact match to detect whether each element appeared in the generated response. In the meantime we evaluate the responses on an element level instead of on a token level. Here’s our formula for calculating F1-score:
> $F_1 = \frac{2TP}{2TP+FP+FN}$, where TP(true positive) is when LLMs correctly identify omitted elements, FP(false positive) is when LLMs identify non-omitted elements, and FN(false negative) is when LLM fail to identify omitted elements.
>
> W5: We appreciate the reviewer for pointing out that our work can be practically useful for relating to LLMs’ failure on certain downstream tasks. Examples include:
> - LLM-as-a-judge: LLMs are likely to fail at grading students’ recitations if they cannot effectively identify omissions.
> - Document and code reviewing/auditing: automated review of pull requests, legal documents, medical records, or contracts often requires ensuring nothing important is missing, but LLMs may not be able to flag them.
> - Safety and security: LLMs may fail to recognize important pieces of safety instructions that are missing from a manual, or in a prompt that is used for training another LLM.
> We included some related discussions in Section 5 (Line 242), but we agree with the reviewer that adding more such discussions would highlight the utility of AbsenceBench and will make sure to include them in the next revision.
>
> W6: We acknowledge that the current claim regarding attention being the reason behind AbsenceBench’s difficulty may sound like an overclaim. We would like to clarify that the experiment in Section 4.2 is intended as a preliminary exploration rather than a comprehensive explanation for the claim. To provide more support for our claim, we have extended the scale of our analysis by **ninefold** (all three tasks for Claude-3.7-Sonnet, GPT-4.1-mini, and Llama-4-Maverick). The analysis indicates that attention mechanisms might be a bottleneck for solving AbsenceBench.
>
> |Models|Poetry|Numerical Sequences|GitHub PRs|Average|
> |---|---|---|---|---|
> | **Placeholder: None (baseline)** |||||
> |Claude-3.7-Sonnet|73.5|91.4|35.7|66.9|
> |GPT-4.1-mini|30.2|45.0|31.3|35.5|
> |Llama-4-Maverick|32.8|58.7|29.0|40.2|
> |**Placeholder: `<missing line>`**|||||
> |Claude-3.7-Sonnet|87.4 (+18.9%)|97.7 (+6.9%)|64.9 (+81.8%)|83.3 (+24.5%)|
> |GPT-4.1-mini|50.4 (+66.9%)|59.2 (+31.5%)|46.8 (+49.5%)|52.1 (+46.8%)|
> |Llama-4-Maverick|60.7 (+85.0%)|78.9 (+34.4%)|46.8 (+61.4%)|62.1 (+54.5%)|
> |**Placeholder: `__________`** |||||
> |Claude-3.7-Sonnet|85.5 (+16.3%)|97.2 (+6.3%)|61.3 (+71.7%)|81.3 (+21.5%)|
> |GPT-4.1-mini | 45.9 (+52.0%)|57.3 (+27.3%)|36.5 (+16.6%)|46.5 (+31.2%)|
> |Llama-4-Maverick|53.9 (+64.3%)|73.0 (+24.4%)|36.5 (+25.9%)|54.5 (+35.5%)|
>
> As shown in the table, explicitly marking omissions in the modified context using placeholders significantly boost all three models’ performance by 41.9% on average. Based on the observation, we hypothesize that this improvement is due to the attention mechanism effectively anchoring to these placeholders.
> We will revise this claim in the next version to ensure it is not overstated. Additionally, we outline below several other possible reasons for model failure that could be explored in future work:
> - Training data: LLMs are rarely exposed to supervised examples where they must detect missing content. The lack of exposure limits their performance.
> - Limitations in LLM reasoning: LLMs often generate hallucinated responses during reasoning, and therefore might hallucinate about omissions.
>
> F1: We apologize for the typo and will proofread the paper carefully.
>
> F2: We apologize for the confusion. The Poetry subset was designed to have a uniform distribution regarding the number of lines instead of total number of tokens. We will clarify this in the next revision.
>
> F3: This is very insightful. In Figure 3, we show that reasoning models typically generate thinking tokens several times greater than the document length, which further suggests that the models may attempt to reconstruct the original document using thinking tokens as a way to identify omissions. However, it is difficult to identify the underlying failure mode through lexical analysis of the responses. Here’s an example that QwQ-32B failed on numerical sequences, where **3555** is an omitted number from the sequence:
> > Wait, looking at the recited list: after 3551, the next is 3553, then **3555**, 3557.
>
> The model hallucinates on the fact that 3555 is omitted from the recited list. We believe deeper analysis on reasoning failures would be helpful to analyze the failure mode of LLMs on AbsenceBench, which we hope can be addressed in future work.
>
> F4: We apologize for the confusion here. The claim that model performance starts to decline after 15K tokens is not accurate and we will remove it from the next revision. To investigate how context length affects model performance, we added a regression analysis in Section 4.1 across all three tasks using three selected LLMs (Claude-3.7-Sonnet, GPT-4.1-mini, and Llama-4-Maverick). We will present these results and provide precise terminology in the next revision.
>
> F5: We fully agree with the reviewer that generalizability across domains is important for AbsenceBench. The current choices of domains are motivated by real world applications where AbsenceBench matters and meanwhile being diverse and cross-section, spanning from realistic data (poetry, PRs) to synthetic data (numerical sequences); from natural language (poetry), numerical, to code languages (PRs). Since our findings are consistent across all three domains, we believe AbsenceBench can be applied and generalized to other domains where detecting surface-level omissions is important.
>
> Dataset Code Comments
> We will make the source data for the Poetry domain available in our final code release. However, due to licensing restrictions, we are unable to provide the source data for the GitHub PRs domain. Instead, we will include code that allows users to retrieve this data directly via the GitHub API.

---

> > ### Comment · Reviewer_shFv · 2025-08-02
> >
> > Thanks for the detailed feedback which helped allay many concerns around parameter choices, length of the benchmark and its implications. Raising my score.

---

> > > ### Author Response · Authors · 2025-08-03
> > >
> > > Thank you very much for your thoughtful consideration and for taking the time to review our responses in detail. We greatly appreciate your feedback and we are glad that our additional explanations helped address your concerns.

---

### Decision · Program_Chairs · 2025-09-18

**Decision:**

Accept (spotlight)

**Comment:**

This is the metareview that summarizes the reviews and discussions. This paper introduces a new benchmark called AbsenceBench which is designed to evaluate the abilities of LLMs in locating conspicuously missing information from long inputs. All reviewers agree that the problem is important, the dataset is useful, the evaluation is extensive, and the analysis is interesting and insightful. One reviewer pointed out that the significance of the limitations should be demonstrated through experimental evidence, which should be addressed in the final version.

===== FINAL UPDATE FROM DB Track PCs ====

The final decision for this paper has been taken by the program chairs after consultation with the SACs. All Senior Area Chairs have ranked papers according to the feedback from the AC during the review process. We decided to leave the original meta-review to reflect the opinion of the AC in light of the initial discussions with reviewers and SAC.